# Multi-color dual wavelength vat photopolymerization 3D printing via spatially controlled acidity

Kyle C. H. Chin[1], Grant Ovsepyan[1] & Andrew J. Boydston [1,2,3] ✉

Dual wavelength vat photopolymerization (DW-VP) has emerged as a powerful approach to create multimaterial objects. However, only a limited range of properties have been showcased. In this work, we report the 3D printing (3DP) of multi-color objects from a single resin vat using DW-VP. This was accomplished by concurrently curing resin with visible light and modulating local resin color with 365-nm ultraviolet (UV) light. The key advance was to use a photoacid generator (PAG) in combination with pH responsive dyes in the 3DP resins. The specific color is dictated by the extent of reaction, or local acidity in our case, and controlled by the light dosage and pattern of UV light applied. Multi-color object formation was implemented in two-step processes involving first 3DP to set the object structure, followed by UV exposure, as well as single processes that leveraged DW-VP to create a broad range of vibrant colors and patterns.

Additive manufacturing (AM), commonly referred to as 3D printing (3DP), is a powerful method of creating objects from computer-aided design models and provides unprecedented control over object shape and function[1–3]. Although most AM methods focus on printing single materials, there is a growing desire to print multiple materials at once[4–7]. Multimaterial AM systems further expand design freedom by providing spatial control over various properties of interest throughout an object which can give rise to a myriad of useful, interesting, and novel functions. Thus far, several properties have been explored across AM methods such as disparate stiffness[8–11], thermomechanical properties[12,13], conductivity[14], color[15,16], and solubility[17].

The most common multimaterial AM methods are deposition-based systems, where object formation is achieved by directly depositing material to specified locations such as melt material extrusion, material jetting, and direct ink write material extrusion. These AM approaches enable multimaterial construction centered around delivery of materials to the print head[8,18]. In contrast, vat photopolymerization (VP) AM methods build objects through light-patterned curing of a homogeneous liquid from a material reservoir. The homogeneity of the resin vat makes multimaterial VP challenging due to a general inability to deliver varied materials to a prescribed voxel within a single vat. Despite this difficulty, multimaterial VP is attractive due to its high resolution, smooth surface finish, and fast print times.

Most efforts toward multimaterial VP focus on methods to change out the entirety of a resin for a new one[5]. This is predominantly accomplished by either swapping of resin vats[19–21] or with fluidic devices[15,22,23]. However, these approaches can suffer from increased print times, additional manufacturing complexity, resin contamination, a need to design multiple co-compatible resins, and limitations on the diversity of materials and thus properties. Additionally, although exchanging resins can lead to multiple materials within a single layer, it is tedious as a single layer needs to be addressed multiple times (separately for each material). Therefore, these approaches often restrict material changes to individual layers, which limits the freedom in material placement throughout an object (i.e., only z-axis material variation is enabled).

An alternative method for multimaterial VP leverages chemistry-centered solutions that allow for a single homogeneous vat of resin to be used. In this, the chemistry of the cured material is

[1]Department of Chemical and Biological Engineering, University of Wisconsin, Madison, WI 53706, USA. [2]Department of Chemistry, University of Wisconsin, Madison, WI 53706, USA. [3]Department of Materials Science and Engineering, University of Wisconsin, Madison, WI 53706, USA. ✉e-mail: aboydston@wisc.edu

manipulated during the printing process by altering the properties of light (i.e wavelength and/or intensity)[5,24]. Simply by tuning the intensity of light, a range of properties have been achieved using a single resin vat through control of the local degree of monomer conversion[4,25–28]. However, these approaches often require post-processing to set the final properties or result in minimal disparity in properties.

By manipulating light intensity and wavelength simultaneously, additional levels of control can be gained over the printing process[5,29]. The principle for this method leverages photochemical orthogonality within resin design. This strategy allows for distinct chemistries to be controlled by exposure to different wavelengths of light[24,29,30]. Several works have utilized this method toward dual wavelength multimaterial VP[9,10,31,32]. Specifically, Schwartz et al.[9] and Dolinksi et al.[10] dictated radical and cationic photopolymerizations while Rossegger et al.[32] paired thiol-ene click and coumarin photocycloaddition reactions to control the local stiffnesses of an object.

Thus far, controlling the stiffness of a material has been the primary demonstration for multimaterial VP 3DP, which motivated us to consider if we could design resins to target additional properties. One property of interest was altering the color of a 3D printed object using dual wavelength VP. Beyond esthetics, producing multi-color objects is of interest for applications in data storage[18], camouflage[33], and education[34,35]. Previously, Peng. et al. utilized gradient light intensity to create multi-color objects from a single resin vat by spatially controlling the generation of radicals and including an anthraquinone-based dye that, when oxidized, changes color[16]. Although this was an excellent step toward multi-color VP 3DP, we noted that the printing process and color modulation are necessarily coupled. Another approach that was recently reported involved a two-step process in which color modulation was performed after 3D printing[36]. We wondered if we could introduce a method in which multiple wavelengths of light are used simultaneously, with one for color modulation independent of the wavelength (and chemistry) that dictates object formation.

## Results and discussion
### Development of multi-color resins
Inspired by the vibrant color changes obtainable in acid/base titration, we envisioned a dual wavelength approach to creating multi-color objects that relied on controlling the local acidity within a part via the use of photoacid generators (PAGs). Onium salt PAGs undergo photolysis under UV light resulting in the creation of a Brønsted acid[37,38]. The strength of the generated acid depends on the counter ion species and is often designated a superacid due to bulky and weakly-coordinating anions such as hexafluoroantimonate, hexafluorophosphate, or tetrafluoroborate[38]. We wondered if the resulting acid generation could be controlled with light dosage and allow for the protonation of pH-responsive dyes. To explore this, a pH-responsive dye, bromocresol green (BG), was chosen due to its need for only moderate pH changes and its distinct color change from blue to green to yellow (See Supplementary Fig. 1 for chemical structures). UV illumination of a solution of a PAG, triarylsulfonium hexafluorophosphate salts (TAS), BG, and aqueous 1 M NaOH (added to adjust the initial solution acidity) resulted in a distinct change in solution color from blue to green to yellow indicative of a change in solution acidity (Fig. 1A). The final solution color could be dictated by the duration of the light exposure.

Although color change was accomplished in solution, decreased mobility of reactive species within a crosslinked network could inhibit the same result in solid samples. To test this, a resin formulation consisting of polyethylene glycol diacrylate ($M_n = 250$ Da), phenylbis(2,4,6-trimethylbenzoyl)phosphine oxide (BAPO), TAS, BG, and aqueous 3 M NaOH, F1 (Table 1, Supplementary Fig. 1), was cured with white light and then subsequently exposed to increasing dosages of UV light. To our delight, the cured sample began to change from blue to green to yellow (Fig. 1B). To ensure this was due to formation of acid and not a radical-induced side reaction, a similar resin containing no TAS, F2 (Table 1), was treated in the same fashion and the sample's color change was monitored. We observed some shifting of the blue color, however, the color change was qualitatively different and resulted in a muting of the blue color rather than shifting the color to

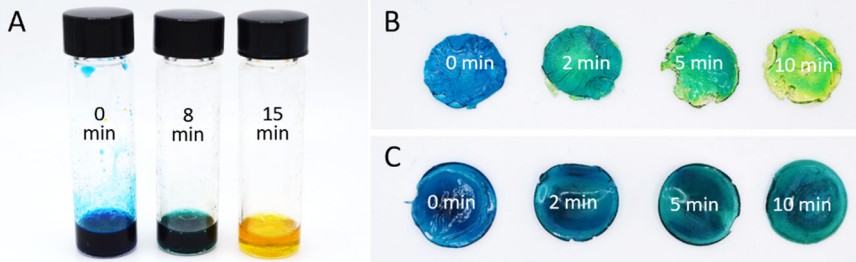

**Fig. 1 | Light facilitated color modulation. A** Solution of TAS, bromocresol green, and aqueous 1 M NaOH after various duration of 10 mW/cm² 365-nm light. **B** F1 that had been cured with white light and then exposed to different durations of 10 mW/cm² 365-nm light. **C** F2 that that had been cured with white light and then exposed to different durations of 10 mW/cm² 365-nm light.

## Table 1 | Investigated resin formulations

| Resin | Dye | PEGDA-250 (wt%)[a] | BAPO (wt%)[b] | TAS (wt%)[b] | Dye (wt%)[b] | 3 M NaOH (wt%)[b] | HQ (wt%)[b] | CQ (wt%)[b] | EDMAB (wt%)[b] |
|---|---|---|---|---|---|---|---|---|---|
| F1 | BG | 100 | 0.4 | 3 | 0.05 | 1 | 0 | 0 | 0 |
| F2 | BG | 100 | 0.4 | 0 | 0.05 | 1 | 0 | 0 | 0 |
| F3 | MR | 100 | 0.4 | 3 | 0.05 | 1 | 0 | 0 | 0 |
| F4 | BG + MR[c] | 100 | 0.4 | 3 | 0.05 | 1 | 0 | 0 | 0 |
| F5 | BG | 100 | 0 | 6 | 0.01 | 1 | 0.2 | 0.4 | 0.4 |

Composition of compounds in each resin formulation. Compounds with their structure, full name, and corresponding acronym can be found in Supplementary Fig. 1.
[a]Considered monomer component.
[b]Weight percent compared to total monomer weight.
[c]Mixture of bromocresol green and methyl red in a ratio of 3:1, 1:1, 1:3 by weight.

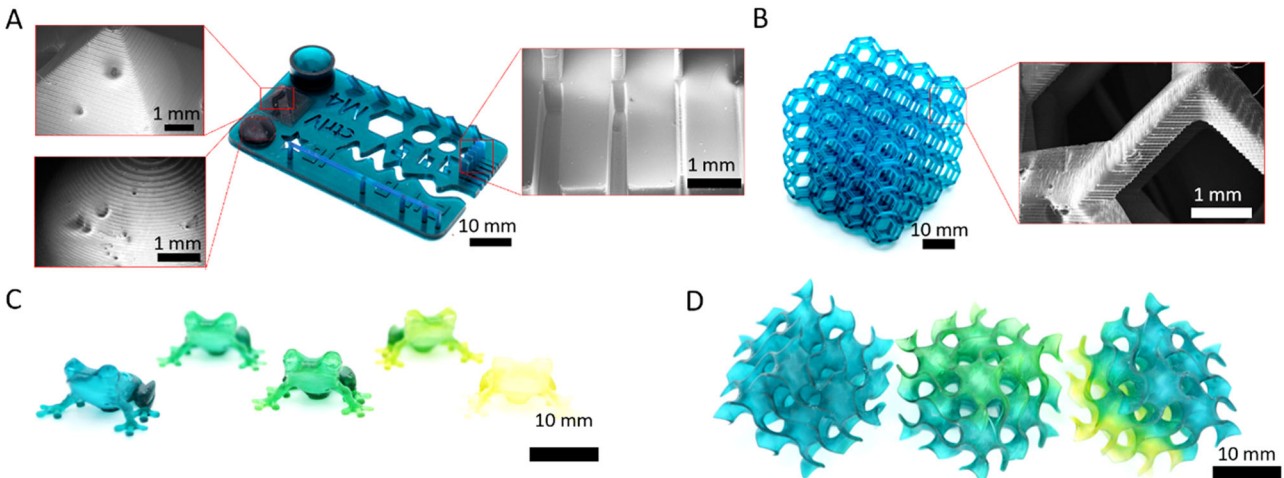

**Fig. 2 | 3D printing of multi-color resin. A** 3D printed sample showcasing the printability and resolution of multi-color resins. SEM image insets showing specific features of the test print. Left insets show a zoomed in image of the pyramid and half dome architectures revealing consistent ~50 μm layers and the right inset reveals feature size resolutions (~100 μm) of 3D printed positive and negative pillars. **B** Complex lattice with SEM inset of a single beam within a unit cell of the lattice. **C** 3D printed frogs using F1 resin and their final color after being altered by 365 nm light for 0, 2, 5, 7, and 15 min as you go from left to right. **D** 3D printed gyroids with their color altered by 365 nm light and physical masks.

green and yellow (Fig. 1C). The change in blue color is consistent with prior reports of BG degradation due to UV photolysis[39,40]. In our studies, we quickly focused on PEGDA-based resin systems given that PEGDA is known to be soluble in water as well as many organic solvents, and is compatible with a range of ionic, hydrophobic, and hydrophilic compounds. Notably, when we briefly examined nonpolar monomers isobornyl acrylate and butyl acrylate we observed solubility limitations from the TAS (Supplementary Fig. 2). We speculate that other photoacid generators could be suitable for use with nonpolar monomers, although this was not explored in our study.

We then explored 3DP of F1 using an Elegoo Mars 3 LCD printer equipped with a single 405-nm light source. With 12 s layer times and 50-μm layer thicknesses (Supplementary Table 1), objects were printed with good resolution and complexity (Fig. 2A–D, Supplementary Fig. 3, Supplementary Fig. 4). Line slits as small as 100 μm, walls as thin as 200 μm, sharp features, and horizontal overhangs were each achieved (Fig. 2A, Supplementary Fig. 4). Also, complex objects could be printed such as lattices, frogs, and gyroids (Fig. 2A–D). The bulk colors of these final objects could be modulated by exposure to UV light after printing (Fig. 2C). In this way, multi-color objects were achieved from a single resin vat. Using physical masks or applying projections results in spatial patterning of objects (Fig. 2D). For thicker objects, color change was less readily accessible due to limitations in light penetration governed by the Beer-Lambert law. To probe this, 10 by 5 by 1 mm samples were illuminated along their top, and the color change recorded as a function of depth (Supplementary Figs. 5, 6). This showed that only shallow penetration of light occurs and color alteration deeper into the sample only takes place after complete discoloration of the outside layers. This is a limitation in the sequential approach to color modulation for thick (>1 mm) object architectures, yet it is encouraging for VP 3DP since unwanted color modulation into already printed layers can be avoided.

We also investigated the effect of light wavelength on the color change of 200-μm thick printed cylinders by exposing printed parts to either 365-, 405-, or 455-nm light (Supplementary Fig. 7). Use of 365-nm light produced the most drastic color change of the wavelengths tested for samples exposed to increasing light dosages. Use of 405-nm light evoked moderate color change but had a sluggish response, only beginning to show small signs of color change after 5 min and 456-nm light exhibited no effect on the color of printed objects even after long exposures. This aligns well with the absorption profile of TAS, which

has negligible absorption of light beyond 390 nm (Supplementary Fig. 8). Importantly, this showcases the ability to decouple the color change from resin curing to still allow postcuring of objects, which can be crucial for setting the final properties of printed objects.

We characterized two important components of the environmental stability of the parts: color stability upon exposure to light, and color leaching when submerged in water. In ambient light without any measures to preclude color changes, we noticed the color of parts gradually evolved toward the color associated with their most acidic form (Supplementary Fig. 9). This effect seemed to be due to ambient light as parts stored in the dark retained their color (Supplementary Fig. 9). Over six weeks of color tracking, the blue and green pixel values stayed rather steady while the red pixel values increased more significantly as the samples changed from appearing blue to green to yellow (Supplementary Fig. 10). To mitigate this, samples were printed from F1 with 0.2 wt% avobenzone, a common UV absorber. Avobenzone has a strong absorption within the UVA region (Supplementary Fig. 8). When these samples were exposed to 365-nm light, no color change was observed which was determined by tracking the RGB values of samples, compared to the distinct color change of F1 (Supplementary Fig. 11). The color stability improved with the addition of the UV blocker, albeit at the expense of rapid color modification during 3DP. We speculate that a UV coating applied after printing and designed to absorb UV-A and UV-B light could solve this issue when a part is made ready for a specific application. Additionally, we investigated color change due to leaching by submerging samples printed from F1 into a solution of water and monitoring the sample color change visually as well as leached pH dye by UV-Vis spectroscopy of the water. No color change was observed (Supplementary Fig. 12) and nearly no detectable leaching of BG into the surrounding water was seen in the absorption spectra over the four weeks (Supplementary Fig. 13). This is attributed to the highly crosslinked nature of the resulting parts which limits the swelling and chance for diffusion of species out of the sample. Notably, other factors that impact swelling, such as affinity of polymer network to solvent, could affect the leaching dynamics of printed objects.

We next investigated accessing a broader range of colors. To do this, we selected methyl red (MR) due to its color change under similar pH ranges as bromocresol green. MR undergoes a distinct transition from yellow to orange to red upon exposure to a more acidic environment. We were able to simply replace MR for BG at the same weight

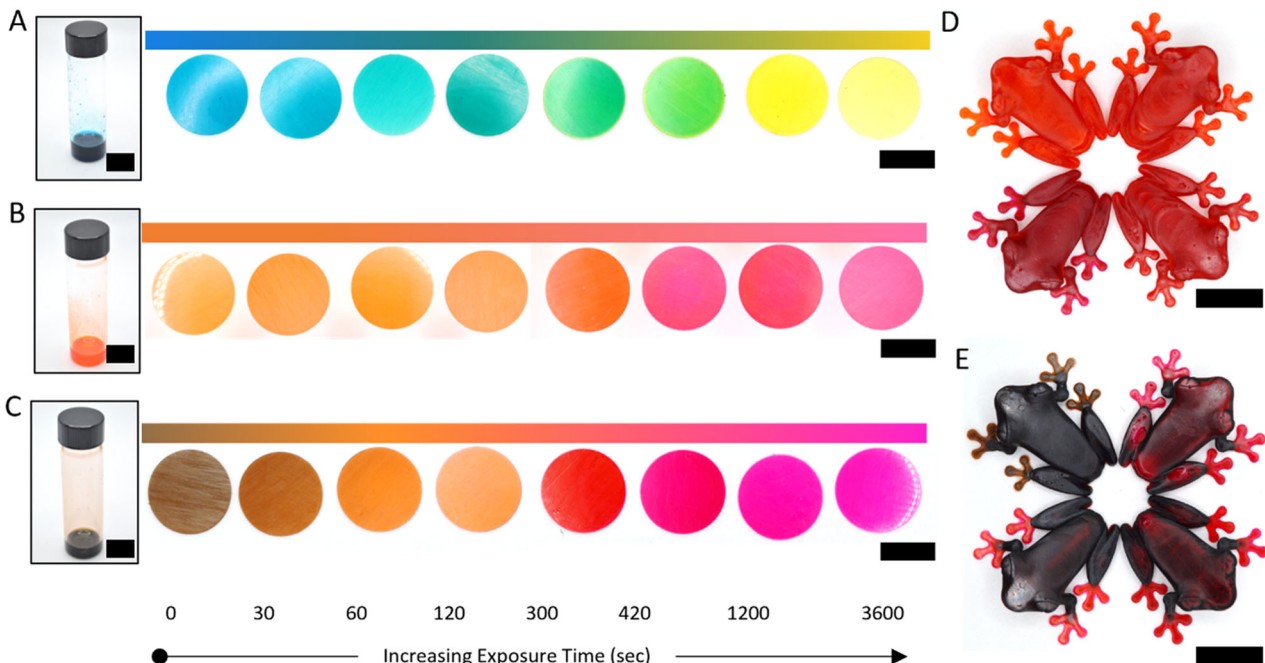

**Fig. 3 | Exploration of color flexibility. A** F1 resin next to printed samples exposed to various exposures of 10 mW/cm² 365-nm light. **B** F3 resin next to printed samples exposed to various exposures of 10 mW/cm² 365-nm light. **C** F4 resin with 1:1 ratio BG and MR next to printed samples exposed to various exposures of 10 mW/cm² 365-nm light. **D** 3D printed frogs using F3 resin and their final color after being altered by 365-nm light for 0, 2, 7, and 15 min (clockwise starting from top left). **E** 3D printed frogs from F4 resin with 1:1 ratio BG and MR and their final colors after being altered by 365-nm light for 0, 2, 7, and 15 min (clockwise starting from top left). All scale bars represent 10 mm.

percent to create F3 (Table 1). The print parameters for F3 were determined iteratively and, overall, the addition of MR had little impact on printing parameters or resolution compared to F1 (Supplementary Table 1, Supplementary Fig. 14). Sample prints along with post print exposure to varying dosages of 365-nm light led to a spectrum of vibrant colors ranging from orange to red, which was distinct compared to the colors accessed from F1 (Fig. 3A, B, D). Conveniently, by mixing BG and MR dyes (F4, Table 1), new and distinct color combinations were possible (Fig. 3C, E) and by controlling the ratio of BG and MR, such as from ratios of 3:1, 1:1, and 1:3 BG:MR, the color palettes could vary (Fig. 3E, Supplementary Fig. 15). These capabilities highlight the flexibility of this approach toward designing resins that can produce a multitude of colors.

## Grayscale patterning

We next wanted to determine if we could recreate patterns of our choice by controlling the local dosage of light in specified regions. To do this, we created a platform for generating grayscale projections to create target patterns. In this approach, an experiment is conducted to determine the color palette of a resin, which provides data to relate grayscale pixel value (0–255) to the resulting color that can be generated by a given resin. With this information, one can move on to transitioning a desired colored image into a grayscale projection that can be used to pattern the target sample. Ideally, desired patterns would be designed with the available color palette in mind, therefore providing simple conversion from the designed projection to the necessary grayscale image using the determined calibration data. However, manipulating any image into a reduced color image that includes only the available colors can be accomplished via minimum variance quantization. The reduced color image can then be translated back into grayscale through the acquired conversion system giving a final projection suitable for printing individual layers. A flowchart showing this process can be found in Supplementary Fig. 16.

We tested our workflow for recreation of target multi-color patterns using the F1 resin formulation. We determined the available colors using a 500-µm thick square 3D printed with uniform light exposure. That sample square was then patterned onto with the grayscale calibration pattern consisting of pixel intensities ranging from 0 to 255 (Fig. 4A). The resulting sample, now displaying a range of colors, was photographed and the available color information extracted (Fig. 4B). We then used MATLAB to extract the colormaps from the grayscale pattern and the swath of available colors, which provided information for translating between the two. Next, we prepared an image of a camoflage pattern (Fig. 4C). The sample image was fed into MATLAB and went through minimum variance quantization to create a reduced color image (Fig. 4D). In our experimental case, the camo pattern was generated in MATLAB using the colors extracted from F1's available colors and therefore needed no alteration. The reduced color image was then translated into grayscale (Fig. 4E) using our obtained translational information and, finally, this could be used to pattern an object with our desired pattern (Fig. 4F). Visually there is good agreement between the target reduced color image and the patterned object. Developing resins with a specific color palette in mind shows promise toward inspiring formulations for coatings, paints, or 3D printing resins geared specifically for creating camouflage materials or tailoring resins to prepare patterns with specified color palettes.

## Dual wavelength multi-color printing

We next took aim at a single process dual wavelength VP 3DP approach. To accomplish this, we employed a custom dual wavelength printer (Supplementary Fig. 17) outfitted with a UV projector (365 nm, max light intensity of ~2 mW/cm²) and a visible light projector (max light intensity of ~200 klux). The printer was operated using MATLAB with Psychtoolbox[41] to control the individual projectors and serial commands to control the z-axis stepper motor. The visible projections were aligned to the UV projector by first undergoing a keystone correction to straighten the image to the projection plane and then

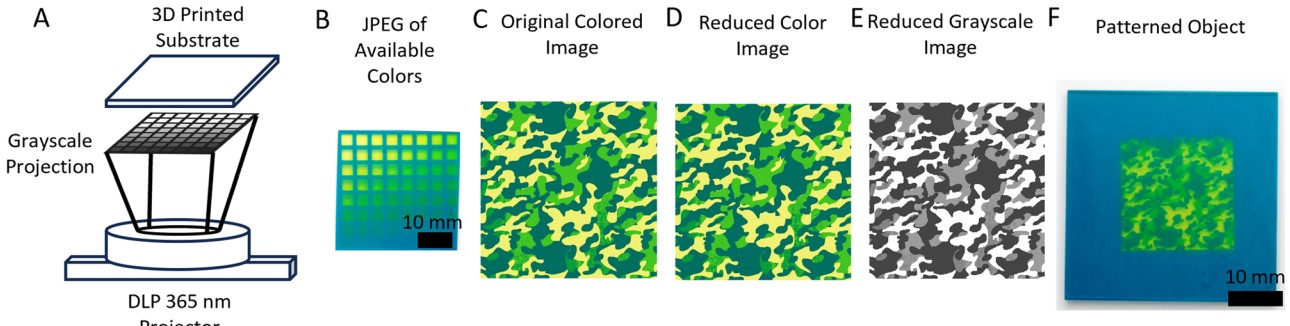

**Fig. 4 | Object patterning using grayscale. A** Patterning a 3D printed substrate with grayscale values (0–255). **B** Photograph of patterned substrate showing the colors obtainable with F1. **C** Image that will be replicated. Camouflage pattern designed with the color palette of F1. **D** Reduced color image of the original target image. **E** Reduced color images translated to the grayscale version for patterning of a substrate. **F** Patterned substrate using the reduced grayscale images as projections for camouflage pattern using F1 resin.

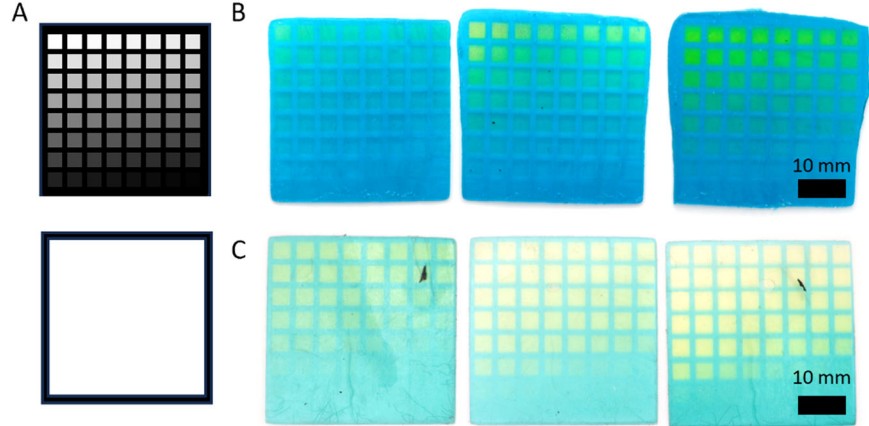

**Fig. 5 | Comparison of DW-VP 3D printing of multi-color resin with different photoinitiation systems. A** UV (top) and visible (bottom) image slices used to determine necessary UV times to express full color range of resins. **B** 3D printed sample printed from F1 resin using 4-sec visible layer cure time and 60- (left), 120- (middle), and 180-sec (right) UV layer time. **C** 3D printed sample printed from F5 resin using 25 sec visible layer cure time and 60- (left), 120- (middle), and 180-s (right) UV layer time.

applying a geometric transformation to each image from the visible projection area to the UV projection area (Supplementary Fig. 18). Notably, print parameters for the visible projector were kept constant for each resin and determined iteratively (Supplementary Table 2). In the future, controlling the visible light dosage within a given layer with grayscale offers additional opportunity to control more properties, such as stiffness, in conjunction with color.

To determine the necessary UV projector time to obtain the full color range, a square was illuminated with white light and overlaid by a gradient intensity of UV light and 1-mm thick samples were printed using a layer thickness of 100 μm and systematically increasing UV layer times (Fig. 5A). Although multi-color objects were printed with multiple colors on a single layer, the printing was overall met with limited success due to excessive UV layer times required to reach the full color change (>3 min/layer) resulting in outgrowth (Fig. 5B). The outgrowth likely stemmed from BAPO's absorbance at 365 nm (Supplementary Fig. 8). To mitigate these issues, the amount of TAS was increased to 6 wt% to aid in faster color change, 0.2 wt% of hydroquinone (HQ) radical inhibitor was added to limit outgrowth, and a type II initiating system consisting of camphorquinone (CQ; $\lambda_{max} = 470$ nm), and ethyl 4-(dimethylamino)benzoate (EDMAB) tertiary amine coinitiator was introduced to reduce radical photoinitiator reactivity at 365 nm (F5, Table 1). Use of CQ had the added benefit of overlapping the absorption profile of the protonated form of BG,

essentially rendering BG as an opaquing agent to the CQ initiator throughout the print and preventing outgrowth at long UV exposure times (Supplementary Fig. 8). Additionally, dye concentration was reduced to better display the color change throughout thicker printed parts. Printing with F5 resulted in faster color change (UV layer times of <3 min/layer) and a discernible reduction in outgrowth (Fig. 5C). Although not explored here, we speculate that shorter layer times could be achieved through use of more efficient PAGs, higher light intensities, or higher concentrations of PAG. However, we found our print parameters to be reasonable to showcase 3DP of these resin formulations. With printability determined, we wondered if the color change was impacting the final properties of 3D printed objects. To test this, we produced blue and yellow ASTM standard type V samples from F5. Quasistatic tensile testing of these samples revealed no statistical difference in ultimate tensile strength, modulus, or elongation to break between the blue and yellow samples printed from the same resin vat (Supplementary Fig. 19).

We next wanted to test our ability to print a range of multi-color structures. To start, we wanted to demonstrate that objects can be printed with multiple colors that would be difficult to produce via subtractive manufacturing. Representative colored CAD models of design files alongside GIFs of the projected layer slices used for dual wavelength 3DP are shown in Supplementary Movie 1. To do this, we printed a multi-color octet truss printed with four different grayscale

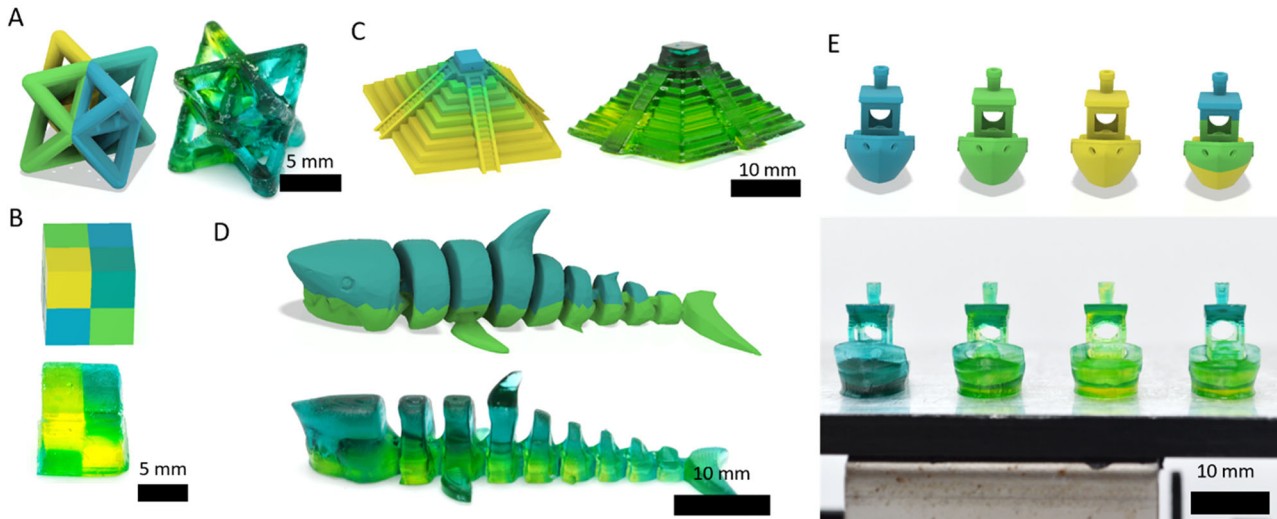

**Fig. 6 | DW-VP 3D printed multi-color 3D printed objects. A** Multi-color octet truss (CAD model (left) printed object (right)). **B** Multi-color cube (CAD model (left) printed object (right)). **C** Gradient multi-color pyramid (CAD model (left) printed object (right)). **D** Two-toned shark (CAD model (left) printed object (right)). **E** Multi-color benchy (CAD model (top) printed objects as printed on build plate (bottom)).

values (Fig. 6A). We also wanted to show the ease of changing colors along the z axis by printing a cube consisting of multiple colors that switch their arrangement halfway up the sample's height as well as a gradient-colored pyramid (Fig. 6B, C). This coloring can also be applied to countershading, a camouflage method used in nature where the topside of an animal is darker colored than the bottom side allowing it to appear flat. Similar to how they appear in nature, a two-toned shark was printed with a darker top half than bottom (Fig. 6D). Additionally, this approach can be used to not only create multi-color objects, but also produce different colored objects from a single resin vat. To showcase this, four benchy models were printed simultaneously on a single build plate resulting in a yellow, green, blue, and multi-color benchy (Fig. 6E).

In conclusion, we have demonstrated a versatile approach for 3DP of multi-color parts via two-step (3DP followed by UV exposure) and single process (dual wavelength) 3DP processes each from single resin vats. Visible light was used to cure the resin and set the final object architecture whereas 365-nm UV light was used to control the color of an object. While traditional radical curing of acrylates was responsible for object formation, color was modulated by controlling the localized acidity within a part. This was accomplished through the inclusion of TAS PAG and various pH-responsive dyes and by controlling the dosage of UV light. Additionally, this approach to multi-color printing was versatile, showing compatibility with multiple pH-responsive dyes that can also be mixed using color mixing rules to achieve an even more diverse range of colors. In the future, we hope to explore additional resin systems that may yield control of new properties in VP using a single resin vat as well as controlling more than a single property through grayscale control of both visible and UV light.

## Methods
### Materials
Bromocresol green (95%), mixed triarylsulfonium hexafluorophosphate salt (50 wt% in propylene carbonate), Phenylbis(2,4,6-trimethylbenzoyl)phosphine oxide (97%), poly(ethylene glycol) diacrylate average with $M_n$ of 250 Da, poly(ethylene glycol) diacrylate average with $M_n$ of 575 Da, poly(ethylene glycol) diacrylate average with $M_n$ of 700 Da, 2-hydroxyethyl acrylate (96 %), Isobornyl acrylate (technical grade), butyl acrylate (≥98%), sodium hydroxide (≥98%), ethyl 4-(dimethylamino)benzoate (≥99%), and camphorquinone (97%)

were purchased from Sigma-Aldrich. Methyl red was purchased from Chem-Cruz. Hydroquinone (99.5%) was purchased from Acros Organics. Avobenzone (97.8%) was purchased from AmBeed. All chemicals were used as received.

### Modifying solution color
A mixture of 3.00 g of TAS, 0.0015 g BG, and 0.05 g 1 M NaOH was prepared. The solution was mixed until homogenous. Contents were then split into three separate vials and a PTFE coated stir bar was added to each. Each vial was exposed to 10 mW/cm² 365 nm light while being stirred at 200 rpm for either 0, 8, or 15 min.

### Resin PTFE mold color change
F1 and F2 resin (0.5 g) was placed into a cylindrical PTFE mold and illuminated with a white light lamp for 15 s which solidified the resin. The cured resin was rinsed with IPA and dried in open air. Each sample was then illuminated with 10 mW/cm² of 365-nm light for either 0, 2, 5, or 10 min.

### Resin preparation
Resin components were mixed in amber borosilicate vials with PTFE-lined caps. Each material was added sequentially (order of addition not important) in the specified amounts. After mixing, the resin was stirred in the dark for 24 h allowing for all components to dissolve.

### Commercial 3D printing
3D printing was done on a commercial Elegoo Mars 3 printer. To print, 3D object files were added to Lychee Slicer and arranged to fit on the build plate. The print profile for each resin was determined iteratively and can be found in Supplementary Table 1. The build plate was levelled before each print. Resin was added to the vat and the print started. After printing, the printed object was rinsed with IPA and separated from the build plate. The parts were then submerged and rinsed in IPA for 5 min to remove uncured resin from the print surface before being postcured with 10 mW/cm² 405-nm light for 1 min.

### UV-Vis spectroscopy
UV-Vis spectroscopy was performed on a Cary 60 UV-Vis spectrophotometer with quartz vials with a 1 cm pathlength at ambient pressure and temperature. A range from 250 to 800 nm was scanned.

## Penetration depth determination

Samples were 3D printed with dimensions of 10 mm × 5 mm × 1 mm (h-l-t). The sides and bottom of each sample were covered using black electrical tape leaving just the top face of the sample exposed. The top of the sample was exposed to 50 mW/cm$^2$ for times from 0 to 1800 sec. The tape was then removed, and the samples photographed (Supplementary Fig. 5). The photos were fed into MALTAB and the color along each height percent of a sample averaged and plotted (Supplementary Fig. 6).

## RGB tracking

Images for RGB tracking were taken with a Nikon D3200 DSLR camera with aperture set to 5.6, shutter speed 1/60, ISO 100, and fine resolution (6016 × 4000). The white balance and focus were set to auto and the camera was 20 cm away from the samples. To ensure consistency between images, a red, green, and blue color swatch was placed in each photo and RGB values extracted to ensure RGB values remained comparable (Supplementary Fig. 10). RGB values for the images were extracted from each sample by taking the average pixel value for ten points on each sample for three replicates.

## Scanning electron microscopy

Samples were sputter-coated using the Leica EM ACE600 with a 10-nm layer of gold. SEM was subsequently conducted on a Zeiss Gemini SEM 450 at an accelerating voltage of 5 kV.

## Multiwavelength testing

Samples were 3D printed that were 500 μm thick and 15 mm in diameter. Either a 365- or 457-nm Kessil lamp or 405-nm projector was positioned to deliver a 10 mW/cm$^2$ light intensity onto a given sample. Intensity was measured using a light meter with the corresponding detector. Samples were then placed under the light for various amounts of time and the resulting colors were captured with a Nikon D3200 camera.

## Photopatterning

Photopatterning was done with a PDC05-5 projector with 365-nm light purchased from Xiamen Zhisen Electro. Equip. Co., LTD that could reach 2 mW/cm$^2$ at the print surface. 40 mm × 40 mm × 0.5 mm (w-l-t) 3D printed sample substrates were placed above the projector and the grayscale calibration pattern displayed for 3 h. After 3 h, the sample was removed and photographed using a Nikon D3200 camera. The image was then fed into MATLAB where the grayscale counterpart was generated. The grayscale image was then projected onto a new sample for 3 h. Ater 3 h the substrate was removed from the projection area now displaying the desired pattern.

## Dual wavelength 3D printing

Dual wavelength 3D printing was accomplished on a custom 3D printer equipped with a Acer X152H visible light projector and a PDC05-5 365 nm light projector purchased from Xiamen Zhisen Electro. Equip. Co., LTD. Print files were prepared in MALTAB. MATLAB was also used to run the printer by controlling projectors via Psychtoolbox[41] and sending G code commands via the serial port.

## Tensile testing

Tensile tests were conducted on a MTS Criterion® Electromechanical Testing System using ASTM D638 type V specimen. The strain rate was set to 10 mm/min. Tensile strain was measured through video analysis using MATLAB by measuring the distance between a series of lines that were drawn on the dogbones in the gauge region prior to testing. Strain was calculated based on the change in distance between the lines on the dogbones. Stress-strain curves were obtained by correlating stress and strain values.

## Measuring light sources emission spectra

General emission spectra for each light source were recorded using a Flame Miniature Spectrometer by Ocean Optics (Supplementary Figs. 21–27). To do this, the spectrometer was placed into a dark enclosure to reduce background noise. The light source of interest was then positioned in front of the light input slit and positioned far enough away so that a reasonable signal to noise ratio was reached at which the signal also did not overload the detector. The spectra were then recorded for each light source and data normalized to the wavelength with the maximum intensity value.

## Photoacid generators (PAGs) solubility tests in different monomers

A solution of 3 wt% TAS in each monomer was prepared in a borosilicate vial. The solution was then lightly perturbed using a vortexer every hour for four hours and dissolution observed. The solutions were left overnight and resulting solubility recorded.

## Data availability

The data that support the findings of this study are available from the authors upon request.

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

## Acknowledgements

A.J.B. acknowledges partial financial support from the Yamamoto Family, the Office of the Vice Chancellor for Research and Graduate Education at the University of Wisconsin-Madison with funding from the Wisconsin Alumni Research Foundation, as well as the Army Research Office (W911NF-20-2-0182-P00005-(76555-EG-MUR) and Office of Naval Research (N00014-23-1-2499). K.C.H.C. would like to thank Jacob Zeuske from the Wisconsin Structures and Materials Testing Laboratory for tensile testing assistance, Blaise Thompson for guidance on constructing our dual wavelength printer, and Rachel Tritt and Xinyu Miao for photography assistance. The authors gratefully acknowledge the use of facilities and instrumentation at the UW-Madison Wisconsin Centers for Nanoscale Technology (wcnt.wisc.edu) partially supported by the NSF through the University of Wisconsin Materials Research Science and Engineering Center (DMR-1720415). This work is also supported by a National Science Foundation Graduate Research Fellowship under Grant No. DGE-1747503 to K.C.H.C. Any opinion, findings, and conclusions or recommendations expressed in this material are those of the authors(s) and do not necessarily reflect the views of the National Science Foundation.

## Author contributions

K.C.H.C. performed the experimental studies, data analysis, and writing. G.O. helped perform experimental studies and projector alignment. A.J.B. supervised the work.

## Competing interests

The authors declare no competing interests.
