## [Peer Review File · Nature Communications]

REVIEWER COMMENTS

Reviewer #1 (Remarks to the Author):

In this manuscript, Boydston and coworkers leverage their expertise in dual wavelength vat photopolymerization in a clever manner that enables the fabrication of multicolored objects. This is accomplished by tuning the release of photoacid in the presence of pH sensitive dyes using a short wavelength of light (UV) while simultaneously using a longer wavelength of light (visible) to selectively cure a rigid acrylic. They systematically demonstrate the ability to finely tune color across the entire visible gamut and to do so with gradient control, while also creating high resolution 3D objects (lattices, horizontal overhangs, etc), and color fastness upon adding UV absorbers. The ability to control color in vat photopolymerizations with this level of control is unprecedented, going well beyond what was demonstrated using grayscale methods (Ref 16) and it will thus enable the production of designer consumer products along with objects that provide useful color guides for educational purposes. Overall, the manuscript is comprehensive and frankly a joy to read. Thus, the work is recommended for publication in Nature Communications pending minor revisions. Additional comments/questions to the authors are provided below.

Comments/questions:

- 1) The authors are encouraged to include the emission profiles of all light sources used, including from the projectors and from the external LEDs used to perform the control measurements.
- 2) The authors are encouraged to include the following recent references in their introduction regarding dual wavelength 3D printing (and the final one being grayscale 3D printing):
 - a. <https://doi.org/10.1021/jacs.3c09567> (Ehrmann and Barner-Kowollik, JACS, 2023, 145, 24438)
 - b. <https://doi.org/10.1016/j.matt.2021.03.021> (Page and co., Matter, 2021, 4, 2172)
 - c. <https://doi.org/10.1021/jacs.1c09523> (Fors and co., 2021, 143, 21200)
 - d. <https://doi.org/10.1016/j.matt.2023.05.040> (Huang, Wallin, and co., Matter, 2023, 6, 2419)
- 3) In the introduction the authors state that “exchanging vats can restrict material changes to individual layers, which limits the freedom in material placement throughout an object”, however this is not necessarily accurate. It is possible to have lateral material variation, but it takes considerably longer time as a single layer needs to be created from both vats and there can be interfacial adhesion issues that arise. The authors are encouraged to modify this statement to account for this.
- 4) In addition to being useful for data storage and camouflage, the ability to recreate natural color in structures has potential for educational purposes as well. The authors may consider adding this point to their introduction to further support the importance of their capability.

5) On page 5 the authors note that the lack of leaching was attributed to the highly crosslinked nature of the parts. It is also likely in-part due to the hydrophobicity of the resin used (PEGDA-250 is a short-chain, oligoethylene glycol (~3-4 repeat units?) and thus not very hydrophilic). Even for a lightly crosslinked print, if it is hydrophobic the lack of swelling in water would prevent leaching. The authors are recommended to note this point as well.

6) On page 8 the word “markable” should I think be “marked”

7) The authors are encouraged to include additional details on their custom dual wavelength 3D printing setup so that others can reproduce the work. For example, how are the projectors combined? What does the temperature controller do (“C” in Figure S15B)? What are the max intensity values for both projectors?

8) The authors are commended on their detailed description of RGB tracking! As a minor note, it currently says “Figure SX” in this description on page 10 of the manuscript. Upon revision the authors should fix this with the correct reference to the SI figure.

9) Probably unimportant query: Is it “Greyscale” or “Grayscale”? I suppose Nature Comm. prefers “greyscale” as used by the authors. But then, it should be colour, not color!

10) Although not necessary for publication here – I think that including a white particle suspension of some sort would really make the surface color pop! I am not sure what is used industrially, but something that is white and not transparent such that light is scattered off the surface would allow for colors in otherwise semi-transparent 3D objects to not be altered by underlying layers (e.g., https://www.amazon.com/ANYCUBIC-UV-Curing-Precision-Excellent-Fluidity/dp/B07G34K22M/ref=sr_1_1?hvadid=604599072686&hvdev=c&hvlocphy=9028287&hvnetw=g&hvqmt=e&hvrnd=432315105976145787&hvtargid=kwd-1645915288475&hydadcr=20106_13297582&keywords=white%2Bresin%2Bfor%2B3d%2Bprinting&qid=1701618843&sr=8-1&th=1).

Reviewer #2 (Remarks to the Author):

This article by Boydston and coworkers reports an original and elegant approach to fabricate multicolor materials, by employing a combination of cationic photoinitiators and pH dyes. First, I would like to congratulate the authors for the smart approach employed. On the other hand, my concerns are mostly related to the impact: the idea is clever, but has it an important application? I’m afraid to say that reading the paper I was thinking that the approach proposed could be great for hobbyist use, but without clear practical applications. In fact, pH indicators were already fabricated via DLP, as reported in literature (eg doi: 10.1021/acsabm.2c00388), including systems able to evaluate ionic force not in aqueous media (eg DOI: 10.1039/d2py01593e). In this case, the change of color is an in-situ generation of protons, so I can say that sensor use cannot be considered.

Obviously, here the scope is just for aesthetic purposes, which is nice but not so impactful. The authors envisaged applications in camouflage and demonstrated the possibility of obtaining complex colors on a single pattern or multicolor in a single objects, but it was also demonstrated that the change of colors occurs spontaneously, over the time, due to environmental driven generation of photocations. The lack of time stability unfortunately decreases the appeal of the envisaged approach.

Related to the complexity of the structures obtained, it was quite expected that 3D printed objects obtained from PEGDA 250 were complex and with high definition, due to the well-known characteristics of this resin. Similarly, due to the low hydrophilicity of this matrix and the tight polymeric network, it was also expected that dye release in water would have been low.

Uv-vis spectra of the resins are missing, and on the final color I also suspect a contribution from CQ at the beginning, which hopefully degrades during UV exposure.

Based on this considerations, my suggestion is to transfer the manuscript to a Journal more focused on scientific soundness than on impact .

Reviewer #3 (Remarks to the Author):

The manuscript under consideration presents a study on the use of dual-wavelength vat polymerization for achieving multi-color prints. The authors utilize the acidity change induced by photoacid generators during photolysis under UV light, in conjunction with a pH-sensitive dye, to demonstrate color change. The paper effectively demonstrates the approach, highlights its limitations, and suggests potential improvements.

While the authors have clearly demonstrated the methodology using PEGDA as the chosen monomer, it would be beneficial if they could expand their investigation to include other monomers. This would help establish the broader applicability of the approach. As PEGDA is known to be soluble in both water and many organic monomers/solvents, its selection may not fully represent the versatility of the technique. A broader range of monomers would provide a more comprehensive assessment.

Below are my detailed comments:

1. Page 2, authors mentioned "Beyond aesthetics, producing multi-color objects is of interest for applications in data storage and camouflage.", can you clarify applications in data storage and camouflage, are they specific to 3D printed multi-color parts, or multi-color parts in general?
2. In Figure 1B, can authors address the color variations with the same sample?
3. In Table 1, c is missing in the Table.
4. In Table 1, authors listed F4 as the only example of mixing pH-sensitive dyes with a weight ratio of 1:1. Have authors tried other compositions, would that affect color change as shown in Figure 3C?

5. Have authors tested the solubility of photoacid generators and pH-sensitive dyes in the monomers, would that limit the choice of monomers that this approach can be used?
6. Figure 2A, it's not clear what is being looked at in the SEM images. Maybe consider optical microscope?
7. Can authors provide information on the exposure time required to achieve the corresponding color change in Figure 2C?
8. In Page 4 and 5, authors demonstrated the color stability upon exposure to light and when submerged in water. Do other post processing steps (i.e., rinsing with solvents) interfere with color change?
9. Specify how the color difference between the frogs in Figure 3D and E were achieved by including exposure times.
10. Figure 4A, clarify the objects shown in the figure.
11. In Experimental section, SI Table and Figure numbers are missing throughout the section.
12. Reference 5 and 19 are the same.

Dear Reviewers,

With your helpful suggestions, we were able to improve the manuscript and are eager to move forward in the process. Below, please see the comments we received as well as our response and actions. In some cases, we have reposted the changes here for your convenience.

Reviewer #1 (Remarks to the Author):

In this manuscript, Boydston and coworkers leverage their expertise in dual wavelength vat photopolymerization in a clever manner that enables the fabrication of multicolored objects. This is accomplished by tuning the release of photoacid in the presence of pH sensitive dyes using a short wavelength of light (UV) while simultaneously using a longer wavelength of light (visible) to selectively cure a rigid acrylic. They systematically demonstrate the ability to finely tune color across the entire visible gamut and to do so with gradient control, while also creating high resolution 3D objects (lattices, horizontal overhangs, etc), and color fastness upon adding UV absorbers. The ability to control color in vat photopolymerizations with this level of control is unprecedented, going well beyond what was demonstrated using grayscale methods (Ref 16) and it will thus enable the production of designer consumer products along with objects that provide useful color guides for educational purposes. Overall, the manuscript is comprehensive and frankly a joy to read. Thus, the work is recommended for publication in Nature Communications pending minor revisions. Additional comments/questions to the authors are provided below.

We thank the reviewer for their time and expertise.

Comments/questions:

1) The authors are encouraged to include the emission profiles of all light sources used, including from the projectors and from the external LEDs used to perform the control measurements.

Normalized emission profiles for each light source have now been reported in the SI (Figure S21 (A-G)) and the method for analyzing emission added to the experimental section. Each has been recreated here for ease of viewing. The new text reads:

“Measuring Light Sources Emission Spectra: General emission spectra for each light source were recorded using a Flame Miniature Spectrometer by Ocean Optics (Figure S21). To do this, the spectrometer was placed into a dark enclosure to reduce background noise. The light source of interest was then positioned in front of the light input slit and positioned far enough

away so that a reasonable signal to noise was reached with the signal also did not overload the detector. The spectra were then recorded for each light source and data normalized to the wavelength with the maximum intensity value.”

Figure S21A) Normalized emission spectra for the white light lamp used to cure objects in PTFE molds.

Figure S21B) Normalized emission spectra for 365 nm Kessil lamp. The wavelength with max absorbance was found to be 369 nm.

Figure S21C) Normalized emission spectra for 456 nm Kessil lamp. The wavelength with max absorbance was found to be 457 nm.

Figure S21D) Normalized emission spectra for 405 nm light source. The wavelength with max absorbance was found to be 404 nm.

Figure S21E) Normalized emission spectra for 365 nm projector. The wavelength with max absorbance was found to be 373 nm.

Figure S21F) Normalized emission spectra for the white light projector.

Figure S21G) Normalized emission spectra for Elegoo Mars 3 LCD 3D printer.

2) The authors are encouraged to include the following recent references in their introduction regarding dual wavelength 3D printing (and the final one being grayscale 3D printing):

- a. <https://doi.org/10.1021/jacs.3c09567> (Ehrmann and Barner-Kowollik, JACS, 2023, 145, 24438)
- b. <https://doi.org/10.1016/j.matt.2021.03.021> (Page and co., Matter, 2021, 4, 2172)
- c. <https://doi.org/10.1021/jacs.1c09523> (Fors and co., 2021, 143, 21200)
- d. <https://doi.org/10.1016/j.matt.2023.05.040> (Huang, Wallin, and co., Matter, 2023, 6, 2419)

We have added the following references to the manuscript. They appear now as reference 29, 24, 30, and 28, respectively.

3) In the introduction the authors state that “exchanging vats can restrict material changes to individual layers, which limits the freedom in material placement throughout an object”, however this is not necessarily accurate. It is possible to have lateral material variation, but it takes considerably longer time as a single layer needs to be created from both vats and there can be interfacial adhesion issues that arise. The authors are encouraged to modify this statement to account for this.

To make this point clearer we have reworded this section of the introduction to say:

“Additionally, although exchanging resins can lead to multiple materials within a single layer, it is tedious as a single layer needs to be addressed multiple times (separately for each material). Therefore, these approaches often restrict material changes to individual layers, which limits the

freedom in material placement throughout an object (i.e., only z-axis material variation is enabled).”

4) In addition to being useful for data storage and camouflage, the ability to recreate natural color in structures has potential for educational purposes as well. The authors may consider adding this point to their introduction to further support the importance of their capability.

We have now added this additional point into the introduction to further support the importance and use for multicolor 3D printing. We have also added two citations showing the importance of color in an educational setting (ref. 34 and 35).

5) On page 5 the authors note that the lack of leaching was attributed to the highly crosslinked nature of the parts. It is also likely in-part due to the hydrophobicity of the resin used (PEGDA-250 is a short-chain, oligoethylene glycol (~3-4 repeat units?) and thus not very hydrophilic). Even for a lightly crosslinked print, if it is hydrophobic the lack of swelling in water would prevent leaching. The authors are recommended to note this point as well.

We agree that the relatively lower hydrophilicity of short chain PEGDA would also play a role in lack of swelling. Additionally, we agree that choice of solvent and its affinity to the polymer network would play a role in the leaching phenomena. We have now noted this in the manuscript text accounting for these factors as well. The new text reads:

“Notably, other factors that impact swelling, such as affinity of polymer network to solvent, could affect the leaching dynamics of printed objects.”

6) On page 8 the word “markable” should I think be “marked”

We have adjusted the text accordingly.

7) The authors are encouraged to include additional details on their custom dual wavelength 3D printing setup so that others can reproduce the work. For example, how are the projectors combined? What does the temperature controller do (“C” in Figure S15B)? What are the max intensity values for both projectors?

We have added additional information about the custom dual-wavelength 3D printing setup in the main text and SI. Notably, the temperature controller was unused in this work, but is otherwise useful to either lower the viscosity of resins or increase reaction rates when needed for other multicomponent resin work. It was noted in the figure as it is an element of the printer although it was not needed for color transformation. We have also expanded on the method for combining the projectors in the SI Figure S18. The new text reads:

“To accomplish this, we employed a custom dual wavelength printer (Figure S17) outfitted with a UV projector (365 nm, max light intensity of ~2 mW/cm²) and a visible light projector (max light intensity of ~200 klux).”

“The visible projections were aligned to the UV projector by first undergoing a keystone correction to straighten the image to the projection plan and then applying a geometric

transformation to each image from the visible projection area to the UV projection area (Figure S18).”

Figure S18. Flowchart showing the process used to align the visible and UV projector images. Both hardware and software were used to align images. The hardware alignment involved keystone correction to account for image distortion from projector angle. Then image transformation of each layer slice of the visible image is performed using a calibration by mapping a grid for each projector area.

Calibration Step (only needs to be performed once for a given printer setup):

1. Both projectors show a grid of 493 evenly spaced points across the entire projection area
2. The pixel coordinates for the intersection of the grid lines were selected from top left to bottom right in sequential order.
3. The same was done for the UV projection grid.
4. Each point was saved in an array.

5. Point coordinated of visible projector were stored in points_in array, while corresponding points on UV projector were stored in points_out array.
6. output_dim (resolution of UV projector), points_in (coordinates of clicked points on Visible projected image), and points_out (coordinates of of clicked points on uv projected image) were exported as 493_points.mat file, which is recovered during implementation step.

Implementation (align the larger projection, in our case the visible projector, to the smaller projection, in our case the uv projector)

1. The imwarp function was used to shrink the visible projection down to the UV projection using the point mapping previously identified in the calibration step (493_points.mat).
2. Pad the resized image on all sides using padarray, replicating edge values to fill new pixels. Since the edge values typically correspond to black color, it practically means we are assigning a 'black' color to fill the newly added borders, integrating them seamlessly with the original content of the image.
3. Repeat for all of the projections for a given sliced 3D model.

8) The authors are commended on their detailed description of RGB tracking! As a minor note, it currently says "Figure SX" in this description on page 10 of the manuscript. Upon revision the authors should fix this with the correct reference to the SI figure.

We fixed this mistake and reviewed the manuscript for other typos as well.

9) Probably unimportant query: Is it "Greyscale" or "Grayscale"? I suppose Nature Comm. prefers "greyscale" as used by the authors. But then, it should be colour, not color!

We have found instances in Nature Communications where grayscale and color were used. We have chosen to go that route and updated the document for consistency. We will make any changes the journal suggests.

10) Although not necessary for publication here – I think that including a white particle suspension of some sort would really make the surface color pop! I am not sure what is used industrially, but something that is white and not transparent such that light is scattered off the surface would allow for colors in otherwise semi-transparent 3D objects to not be altered by underlying layers (e.g., https://www.amazon.com/ANYCUBIC-UV-Curing-Precision-Excellent-Fluidity/dp/B07G34K22M/ref=sr_1_1?hvadid=604599072686&hvdev=c&hvlocphy=9028287&hvnetw=g&hvqmt=e&hvrnd=432315105976145787&hvtargid=kwd-1645915288475&hydadcr=20106_13297582&keywords=white%2Bresin%2Bfor%2B3d%2Bprinting&qid=1701618843&sr=8-1&th=1).

This could be a nice avenue for future investigation. A common pigment used in industry is TiO₂, but the absorption profile of this in our system would hinder the PAG. We would like to look into this into the future.

Reviewer #2 (Remarks to the Author):

This article by Boydston and coworkers reports an original and elegant approach to fabricate multicolor materials, by employing a combination of cationic photoinitiators and pH dyes. First, I

would like to congratulate the authors for the smart approach employed. On the other hand, my concerns are mostly related to the impact: the idea is clever, but has it an important application? I'm afraid to say that reading the paper I was thinking that the approach proposed could be great for hobbyist use, but without clear practical applications. In fact, pH indicators were already fabricated via DLP, as reported in literature (eg doi: 10.1021/acsabm.2c00388), including systems able to evaluate ionic force not in aqueous media (eg DOI: 10.1039/d2py01593e). In this case, the change of color is an in-situ generation of protons, so I can say that sensor use cannot be considered.

Obviously, here the scope is just for aesthetic purposes, which is nice but not so impactful. The authors envisaged applications in camouflage and demonstrated the possibility of obtaining complex colors on a single pattern or multicolor in a single objects, but it was also demonstrated that the change of colors occurs spontaneously, over the time, due to environmental driven generation of photocations. The lack of time stability unfortunately decreases the appeal of the envisaged approach.

Related to the complexity of the structures obtained, it was quite expected that 3D printed objects obtained from PEGDA 250 were complex and with high definition, due to the well-known characteristics of this resin. Similarly, due to the low hydrophilicity of this matrix and the tight polymeric network, it was also expected that dye release in water would have been low.

Uv-vis spectra of the resins are missing, and on the final color I also suspect a contribution from CQ at the beginning, which hopefully degrades during UV exposure.

Based on this considerations, my suggestion is to transfer the manuscript to a Journal more focused on scientific soundness than on impact .

We thank the reviewer for their time and expertise. We are glad the reviewer found the science to be sound and the approach elegant. Although one can surely imagine applications beyond what we might speculate upon in this manuscript, we would also argue the general importance of aesthetics in the world. If this is an aspect that holds 3D printing back, as an industry, then publishing methods for solving the problem could have much larger implications.

Reviewer #3 (Remarks to the Author):

The manuscript under consideration presents a study on the use of dual-wavelength vat polymerization for achieving multi-color prints. The authors utilize the acidity change induced by photoacid generators during photolysis under UV light, in conjunction with a pH-sensitive dye, to demonstrate color change. The paper effectively demonstrates the approach, highlights its limitations, and suggests potential improvements.

While the authors have clearly demonstrated the methodology using PEGDA as the chosen monomer, it would be beneficial if they could expand their investigation to include other monomers. This would help establish the broader applicability of the approach. As PEGDA is known to be soluble in both water and many organic monomers/solvents, its selection may not fully represent the versatility of the technique. A broader range of monomers would provide a more comprehensive assessment.

We thank the reviewer for their time and expertise.

Below are my detailed comments:

1. Page 2, authors mentioned “Beyond aesthetics, producing multi-color objects is of interest for applications in data storage and camouflage.”, can you clarify applications in data storage and camouflage, are they specific to 3D printed multi-color parts, or multi-color parts in general?

To clarify, any colored object can be used for camouflage or data storage to store information. However, 3D printed multicolor objects have an added benefit of being able to contain information not only on the surface of the part but also within the part. This would be difficult if not impossible to achieve with other manufacturing approaches. Additionally, this can help to better encode detail or information across the volume of your part, not just on the surface.

2. In Figure 1B, can authors address the color variations with the same sample?

These samples were cured with a rather intense white light lamp and exposed to air at the top. This led to the formation of an uneven surface of final cured parts. The color variation is most likely due to this surface variation as the same resin 3D printed with an even surface resulted in no color variation. Ultimately, a more controlled and refined setup fixed any unevenness we initially observed in the first control experiments.

3. In Table 1, c is missing in the Table.

The c superscript has now been added to the table near the BG+MR in the dye column.

4. In Table 1, authors listed F4 as the only example of mixing pH-sensitive dyes with a weight ratio of 1:1. Have authors tried other compositions, would that affect color change as shown in Figure 3C?

We had not tried other mixing ratios, but have now explored this and added it to the SI Figure S15 and reproduced it below for quick reference. In brief, mixing different ratios of BG and MR substantially altered the color combinations accessible using this approach which adds even more breadth and flexibility. We thank the reviewer for their suggestion.

“Figure S15) A) Resin formulation for F4 made with a 1:3 ratio of BG and MR. To the right of the resin are printed disks exposed to increasing dosages of UV 365-nm light. B) Resin formulation for F4 made with a 3:1 ratio of BG and MR. To the right of the resin are printed disks exposed to increasing dosages of UV 365-nm light.”

Resin	Dye	PEGD A-250 (wt%) ^a	BAPO (wt%) ^b	TAS (wt%) ^b	Dye (wt%) ^b	3 M NaOH (wt%) ^b	Comments
F4	BG+MR	100	0.4	3	0.05	1	Dye was mixed at a ratio of 3:1 BG:MR
F4	BG+MR	100	0.4	3	0.05	1	Dye was mixed at a ratio of 1:3 BG:MR

5. Have authors tested the solubility of photoacid generators and pH-sensitive dyes in the monomers, would that limit the choice of monomers that this approach can be used?

We performed a limited screening based upon the success with PEGDA-based resins. In general, solubility limitations were easy to observe visually and could be established if desired by future researchers. Notably TAS was not soluble in nonpolar monomers, but we speculate other PAGs would address this.

6. Figure 2A, it's not clear what is being looked at in the SEM images. Maybe consider optical microscope?

To make this more clear, additional text was added in the figure caption explaining what the SEM images are looking at. We hope this adds clarity to this portion of the manuscript.

7. Can authors provide information on the exposure time required to achieve the corresponding color change in Figure 2C?

The color changes for each sample have now been added. They were 0, 2, 5, 7, and 15 min respectively from left to right.

8. In Page 4 and 5, authors demonstrated the color stability upon exposure to light and when submerged in water. Do other post processing steps (i.e., rinsing with solvents) interfere with color change?

Notably, all parts were rinsed with IPA to remove uncured material. This did not have an impact on its color as the dye within the part was locked in place. Since this is not a covalent interaction, swelling in compatible solvents would remove color over time. However, short rinses with solvent did not impact the color of the samples that we printed.

9. Specify how the color difference between the frogs in Figure 3D and E were achieved by including exposure times.

The color changes for each sample have now been added. They were 0, 2, 7, and 15 minutes starting from top left going clockwise.

10. Figure 4A, clarify the objects shown in the figure.

The image in Figure 4A is now labeled for better clarity. The bottom object is the DLP 365 nm projector, the middle object is the 0-255 projected greyscale image that will be

patterned onto substrate, and the top object is the printed substrate that you hope to pattern. The figure has been recreated below for convenience.

Figure 4. Depiction of the process of patterning objects with a desired pattern using multi-color resins. A) Patterning a 3D printed substrate with grayscale values (0 – 255). B) Photograph of patterned substrate showing the colors obtainable with F1 (top) and F4 (bottom). C) Image that will be replicated. Camouflage pattern designed with the color palette of F1 (top) and fall leaves image that uses similar colors to F4 (bottom). D) Reduced color image of each of the original target images. E) Reduced color images translated to the grayscale version for patterning of a substrate F) Patterned substrate using the reduced grayscale images as projections for camouflage pattern using F1 resin (top) and fall leaves using F4 resin (bottom).

11. In Experimental section, SI Table and Figure numbers are missing throughout the section.
We fixed this mistake and reviewed the manuscript for other errors as well.

12. Reference 5 and 19 are the same.
We fixed this mistake and reviewed the manuscript for other errors as well.

REVIEWERS' COMMENTS

Reviewer #3 (Remarks to the Author):

In the updated manuscript, the authors have effectively addressed most of the concerns raised previously and have enriched their work with new results. Here are my specific observations and suggestions:

1. In response to the earlier comment 4, the authors have incorporated additional BG:MR ratios (1:3 and 3:1) in Table 1. Consequently, I recommend that the caption of Figure 3C should be updated to explicitly state that the BG:MR ratio of 1:1 was used for the F4 resin.
2. It would be insightful if the authors could discuss the feasibility of extending their methodology to a wider array of monomers, including any potential obstacles that might be encountered in such an endeavor.
3. Considering the relevance of a recent publication (<https://pubs.acs.org/doi/full/10.1021/cbe.3c00088>), which employs a methodology similar to that presented in this manuscript, I suggest the authors provide their perspective on this study.

Dear Reviewers,

With your helpful suggestions, we were able to improve the manuscript and are eager to move forward in the process. Below, please see the comments we received as well as our response and actions. In some cases, we have reposted the changes here for your convenience.

Reviewer #3 (Remarks to the Author):

In the updated manuscript, the authors have effectively addressed most of the concerns raised previously and have enriched their work with new results. Here are my specific observations and suggestions:

1. In response to the earlier comment 4, the authors have incorporated additional BG:MR ratios (1:3 and 3:1) in Table 1. Consequently, I recommend that the caption of Figure 3C should be updated to explicitly state that the BG:MR ratio of 1:1 was used for the F4 resin.

We have updated the Figure caption as suggested.

2. It would be insightful if the authors could discuss the feasibility of extending their methodology to a wider array of monomers, including any potential obstacles that might be encountered in such an endeavor.

We appreciate the suggestion. We included in the revised manuscript some commentary on potential solubility issues. We would prefer to maintain the focus of the paper on the proven capabilities and learning outcomes, without additional speculation on what might arise as the chemistry is changed in future iterations.

3. Considering the relevance of a recent publication (<https://pubs.acs.org/doi/full/10.1021/cbe.3c00088>), which employs a methodology similar to that presented in this manuscript, I suggest the authors provide their perspective on this study.

We thank the reviewer for bringing this publication to our attention. We have now cited the paper in our manuscript and included a concise statement acknowledging the publication as well as a key difference with our current manuscript (two-step processing versus simultaneous 3D printing with color modulation).